# Selective Decontamination of the Digestive Tract to Prevent Postoperative Pneumonia and Anastomotic Leakage after Esophagectomy: A Retrospective Cohort Study

**DOI:** 10.3390/antibiotics10010043

**Published:** 2021-01-05

**Authors:** Robin Janssen, Frans Van Workum, Nikolaj Baranov, Harmen Blok, Jaap ten Oever, Eva Kolwijck, Alma Tostmann, Camiel Rosman, Jeroen Schouten

**Affiliations:** 1Department of Intensive Care Medicine, Radboud Institute for Health Sciences, Radboud University Medical Center, Geert Grooteplein Zuid 10, 6525 GA Nijmegen, The Netherlands; Jeroen.Schouten@radboudumc.nl; 2Radboud Center of Infectious Diseases, Radboud University Medical Center, Geert Grooteplein Zuid 10, 6525 GA Nijmegen, The Netherlands; jaap.tenoever@radboudumc.nl (J.t.O.); alma.tostmann@radboudumc.nl (A.T.); 3Department of Surgery, Radboud University Medical Center, Geert Grooteplein Zuid 10, 6525 GA Nijmegen, The Netherlands; frans.vanworkum@radboudumc.nl (F.V.W.); nikolaj.baranov@radboudumc.nl (N.B.); Camiel.rosman@radboudumc.nl (C.R.); 4Faculty of Medical Sciences, Radboud University, Geert Grooteplein Noord 21, 6525 EZ Nijmegen, The Netherlands; h.a.blok@student.ru.nl; 5Department of Internal Medicine, Radboud University Medical Center, Geert Grooteplein Zuid 10, 6525 GA Nijmegen, The Netherlands; 6Department of Medical Microbiology, Jeroen Bosch Hospital, Henri Dunantstraat 1, 5223 GZ Den Bosch, The Netherlands; e.kolwijck@jbz.nl; 7Unit Hygiene and Infection Prevention, Department of Medical Microbiology, Radboud University Medical Center, Geert Grooteplein, Zuid 10, 6525 GA Nijmegen, The Netherlands

**Keywords:** esophagectomy, postoperative infectious complications, pneumonia, anastomotic leakage, selective decontamination of the digestive tract (SDD)

## Abstract

Infectious complications occur frequently after esophagectomy. Selective decontamination of the digestive tract (SDD) has been shown to reduce postoperative infections and anastomotic leakage in gastrointestinal surgery, but robust evidence for esophageal surgery is lacking. The aim was to evaluate the association between SDD and pneumonia, surgical-site infections (SSIs), anastomotic leakage, and 1-year mortality after esophagectomy. A retrospective cohort study was conducted in patients undergoing Ivor Lewis esophagectomy in four Dutch hospitals between 2012 and 2018. Two hospitals used SDD perioperatively and two did not. SDD consisted of an oral paste and suspension (containing amphotericin B, colistin, and tobramycin). The primary outcomes were 30-day postoperative pneumonia and SSIs. Secondary outcomes were anastomotic leakage and 1-year mortality. Logistic regression analyses were performed to determine the association between SDD and the relevant outcomes (odds ratio (OR)). A total of 496 patients were included, of whom 179 received SDD perioperatively and the other 317 patients did not receive SDD. Patients who received SDD were less likely to develop postoperative pneumonia (20.1% vs. 36.9%, *p* < 0.001) and anastomotic leakage (10.6% vs. 19.9%, *p* = 0.008). Multivariate analysis showed that SDD is an independent protective factor for postoperative pneumonia (OR 0.40, 95% CI 0.23–0.67, *p* < 0.001) and anastomotic leakage (OR 0.46, 95% CI 0.26–0.84, *p* = 0.011). Use of perioperative SDD seems to be associated with a lower risk of pneumonia and anastomotic leakage after esophagectomy.

## 1. Introduction

Esophagectomy is a complex surgical procedure, associated with significant morbidity and mortality rates [1,2]. Most postoperative complications involve infectious complications (10–30%), including pulmonary infections, surgical-site infections (SSIs), and anastomotic leakage [1,2,3,4]. Microbiota, especially of the oropharyngeal and digestive tract, provide an important source of pathogens causing hospital-acquired infections [5,6,7]. Similarly, microorganisms may play a major role in the pathogenesis of anastomotic leakage. Pathogens originating from the gastrointestinal (GI) tract reaching the site of the anastomosis can induce local inflammation with abscess formation, facilitating anastomotic dehiscence and, eventually, anastomotic leakage [8,9,10]. Eliminating these pathogens prior to surgery and during recovery may decrease the risk of infections and anastomotic leakage.

Selective decontamination of the digestive tract (SDD) is a prophylactic antibiotic strategy that aims to prevent infections by reducing GI colonization with aerobic gram-negative rods and yeasts, while preserving anaerobic microbiota [11]. SDD consists of oral, non-absorbable antimicrobial agents, e.g., colistin, tobramycin, and amphotericin [12]. SDD has been shown to improve the outcome in intensive care unit (ICU) patients [11,12,13,14]. In a systematic review, Roos et al. showed that SDD decreased the rate of postoperative infections and anastomotic leakage in GI surgery [15]. Given the high infectious complication risk after esophagectomy, SDD might be a promising strategy to reduce postoperative infections in this category of patients. In esophageal surgery, earlier studies have reported SDD to reduce gram-negative colonization and to decrease postoperative infection rates after esophagectomy [16,17]. However, these two studies included a very limited patient group from a single center, and the surgical technique, perioperative care, and SDD protocols have changed over time. Robust evidence for SDD effectiveness in esophageal cancer surgery is thus lacking.

The aim of this study was to evaluate SDD as a potentially protective strategy for infectious complications (pneumonia and SSIs), anastomotic leakage, and mortality in patients undergoing esophagectomy.

## 2. Materials and Methods

### 2.1. Ethics

The study was conducted in accordance with the Declaration of Helsinki and approved by the research ethics committee of the Radboud university medical center Nijmegen (file number 2018-4296). Local approval was obtained from the medical ethical committees at the participating medical centers, waiving the need for informed consent.

### 2.2. Study Design and Patients

A retrospective cohort study was conducted in patients aged 18 years or older with esophageal carcinoma (pT1-4aNxM0 tumors) who underwent elective Ivor Lewis totally minimally invasive esophagectomy (TMIE) between 2012 and 2018 in the Netherlands. There were no exclusion criteria. Patient data were collected from a quality of care registry of four high-volume esophageal cancer centers and from the electronic patient files. The four hospitals each performed at least 40 cases of Ivor Lewis TMIE per year.

### 2.3. Procedure

All patients underwent Ivor Lewis TMIE, consisting of laparothoracoscopic resection with intrathoracic anastomosis [18]. The four centers all use comparable perioperative protocols based on national guidelines, discuss patients in multidisciplinary teams, and regularly exchange information on perioperative care in the Dutch Upper GI Cancer Group. All the operations were performed in teams of two surgeons. The number of surgeons per hospital specially trained to perform this specific operation differed per hospital: in hospital 1, there were three surgeons; in hospital 2, there were two surgeons; in hospital 3, there were also two surgeons; and in hospital 4, there were three surgeons. Generally, the qualified surgeons operated together; occasionally, however, the surgery was performed by one specially trained surgeon in combination with another general GI-surgeon or a surgeon in training. Intravenous perioperative antimicrobial prophylaxis was given to all patients according to local hospital guidelines, which are based on the guidelines issued by the Dutch Working Party on Antibiotic Policy (SWAB) for perioperative prophylaxis (patients received 2000 mg of cefazolin intravenously and 500 mg of metronidazole intravenously 30 to 60 min prior to surgery, which was repeated if duration of surgery exceeded four hours) [12]. In patients who were expected to be on the ventilator postoperatively for >48 h and/or expected to remain in ICU for >72 h, intravenous Cefotaxime 4 dd 1 g was administered for four days (this constitutes regular care for ICU patients irrespective of their indication for admission according to prevailing SWAB guidelines) [12].

### 2.4. Selective Decontamination of the Digestive Tract (SDD)

SDD is not incorporated in the Dutch National Guideline for esophagectomy. As a result, there is a discrepancy in Dutch centers using perioperative SDD. In this study, we included two hospitals that routinely use perioperative oral SDD (hospital 1 and 4) and two hospitals (hospital 2 and 3) that do not use SDD. The SDD strategy consisted of an oral paste and an oral suspension. The oral paste consists of amphotericin B 2%, colistin 2%, and tobramycin 2%, and was applied to the oropharynx. The oral suspension (solution) contained a mixed solution of amphotericin B (50 mg/mL), colistin sulphate (10 mg/mL), and tobramycin (8 mg/mL), which was taken orally and swallowed. Patients started prophylactic treatment at home three days prior to surgery, taking 10 mL of the oral suspension four times daily. The oropharyngeal paste was started after surgery and continued until extubation, while oral suspension was continued until three days after surgery.

### 2.5. Data and Data Collection

Patient demographic and clinical data included gender, year of birth, body mass index (BMI), chronic use of immunosuppressive agents, American Society of Anesthesiology physical status (ASA classification), Charlson Comorbidity Index (CCI), tumor histology, clinical tumor stage (TNM stage), tumor location, and neoadjuvant treatment. Peroperative variables included hospital of surgery, year of surgery, use of SDD, duration of surgery, number of resected lymph nodes, epidural analgesia, and operation related parameters (blood loss and conversion to open surgery (abdominal, thoracic)). Postoperative variables collected were pneumonia, SSIs, anastomotic leakage, empyema/mediastinitis, chylothorax, length of stay, readmissions within 30 days, reintervention, and mortality. All data were entered into Castor EDC, a licensed and secured online data collection management system for medical research [19].

### 2.6. Outcomes

The primary outcomes were pneumonia and SSIs, occurring within the 30-day postoperative period. Pneumonia was defined according to the revised Uniform Pneumonia Score (rUPS) [4]. SSIs were defined as infections of the incision site or operative space that required antibiotic treatment and/or any form of surgery/drainage, and were classified as abdominal, thoracic wound infections, or jejunostomy-associated wound infections [20]. Secondary outcomes were postoperative anastomotic leakage defined and graded according to Low et al. (2015) [21], as well as all-cause 1-year mortality.

Other outcomes were length of ICU and hospital stay, ICU and hospital readmission within 30 days, and reintervention after primary surgery (surgical, radiological, endoscopic), as well as in hospital, 30-day, and 90-day mortality.

### 2.7. Statistical Analysis

Statistical analysis was performed using SPSS version 25.0 software (IBM Corporation, Armonk, NY, USA). Continuous variables were expressed as the mean (standard deviation) or the median (interquartile range (IQR)), and categorical variables were expressed as frequency (percentage). For continuous variables, Student’s t-test was used for normal distribution, and non-parametric tests for non-normal distribution. Categorical variables were analyzed using Pearson’s chi-square or Fisher’s exact test.

The main association of interest was that between SDD and the primary outcomes (pneumonia and SSIs) and secondary outcomes (anastomotic leakage and 1-year mortality), all measured in odds ratio (OR). The association was first assessed with univariate logistic regression analysis. Multivariate logistic regression was then performed to reduce possible confounding. All variables associated with the dependent variables in univariate analysis (*p* < 0.10), together with variables considered clinically relevant based on the literature and expert opinions, were selected for multivariate logistic regression analysis. To prevent overfitting, the number of variables in the multivariate analysis was restricted by the number of patients with the outcome. Missing data in logistic regression were handled by listwise deletion.

In a post-hoc sensitivity analysis on the effect of SDD on (infectious) complications, we investigated to what extent differences between hospitals (e.g., surgical learning curve or perioperative care) could have influenced our results. We evaluated the occurrence of postoperative (infectious) complications in one hospital (hospital 1), before and after implementation of SDD, and compared this with another hospital where no SDD was used (hospital 2). For this analysis, we used data from 86 additional patients from hospital 1 from the pre-SDD period (2012–2013). Data on patients who underwent Ivor Lewis TMIE without perioperative SDD were unavailable in the other SDD hospital, because no Ivor Lewis TMIE was performed in this hospital before implementation of the SDD pathway.

All tests were two-sided, and a *p*-value of ≤ 0.05 was considered to be statistically significant.

## 3. Results

Patient characteristics, surgical characteristics, postoperative consequences, and postoperative (infectious) complications can be found in Table 1. In total, 496 patients were included in the study, of whom 179 (36.1%) received SDD. The mean age of the study group was 64.9 years (SD 8.4), with 417 participants (84.1%) being male.

Pneumonia was the most common type of infectious complication, affecting 153 (30.8%) patients. Surgical-site infections occurred in 17 (3.4%) patients. A total of 82 (16.5%) patients developed anastomotic leakage. Mediastinitis/empyema and chylothorax occurred in 60 (12.1%) and 48 patients (9.7%) patients, respectively (Table 1).

ASA classification was more favorable in the SDD group, but the average clinical tumor stage was lower in the non-SDD group. More lymph nodes were resected in the non-SDD group (23 vs. 20, p = 0.001). Median blood loss was higher in the SDD group (200 vs. 174 mL, *p* = 0.042) and there was a higher rate of conversion to open surgery in the SDD group (8.4% [15/179] vs. 1.6% [5/317], *p* < 0.001) (Table 1).

Median ICU length of stay and hospital length of stay were both longer in patients using SDD (ICU 4 vs. 1 day, *p* < 0.001; hospital 13 vs. 11 days, *p* < 0.001). Readmission, reintervention, and mortality did not significantly differ between the SDD and non-SDD group (Table 1).

Patients using SDD experienced significantly less pneumonia (20.1% [36/179] vs. 36.9% [117/317], *p* < 0.001) and anastomotic leakage (10.6% [19/179] vs. 19.9% [63/317], *p* = 0.008) compared with patients not using SDD. No statistically significantly differences were found between the SDD and non-SDD group for occurrence of SSIs (5.0% [9/179] vs. 2.5% [8/317], *p* = 0.141) and for 1-year mortality (26.8% [48/179] vs. 20.4% [59/317], *p* = 0.056) (Table 1).

In multivariate analysis, SDD was an independent factor associated with a lower risk of pneumonia (OR 0.40, 95% confidence interval (CI) 0.23–0.67, *p* < 0.001) and anastomotic leakage (OR 0.46, 95% CI 0.26–0.84, *p* = 0.011), but not with a lower risk of 1-year mortality (OR 1.31, 95% CI 0.80–2.14, *p* = 0.278) (Table 2). For details of these analyses, see Appendix A. Multivariate analysis was not performed for SSIs because of the low number of patients who developed an SSI.

The post-hoc sensitivity analysis is shown in Figure 1. When comparing the incidence of postoperative pneumonia (A) and anastomotic leakage (B) in period 1 (2010–February 2014) with period 2 (March 2014–2017) per hospital, significant decreases in pneumonia and anastomotic leakage were observed only in the hospital where SDD was implemented (hospital 1).

## 4. Discussion

This study found that, in patients undergoing Ivor Lewis TMIE, perioperative SDD use was associated with a reduction in postoperative pneumonia and anastomotic leakage. The reduction in these post-operative complications as a result of SDD did not infer a reduction in 1-year mortality.

In accordance with previous studies, postoperative pneumonia appeared to be the most common infectious complication in patients following esophagectomy [22,23,24]. In the literature, percentages ranging from 7.6 to 35.9% are reported [25]. In our study, pneumonia occurred in 30.8% of all patients. SDD markedly decreased the occurrence of postoperative pneumonia (36.9% vs. 20.1%). As SDD has repeatedly been shown to reduce pneumonia rates in ICU patients [11,12,13,14], this may not come as a surprise and its effect has already been suggested in smaller studies in GI surgery [15,16,17]. Swallowing dysfunction and silent tracheobronchial aspiration occur in a significant number of esophagectomy patients in the early post-operative period [26]; therefore, SDD seems to effectively prevent the development of pulmonary infections evolving from aspiration of oropharyngeal microbiota.

Surgical-site infections occurred in only 3.4% of the patients. The fact that all patients underwent totally minimally invasive surgery (which involves smaller incision sites compared with open surgery), undoubtedly accounted for the relatively low percentages of surgical site infections [27,28]. In contrast with the results of earlier studies, and probably because of the low number of patients with SSIs, we did not observe a reduction in SSIs in patients using SDD when compared with patients not using SDD [15,16].

Further, our results suggest that SDD had a protective effect on the occurrence of anastomotic leakage itself. The observed protective effect of SDD on anastomotic leakage (OR 0.46, 95% CI 0.26–0.84, *p* = 0.011) is remarkable, as anastomotic leakage is thought to be primarily influenced by multiple surgical factors, including the learning curve [29]. Although the pathophysiology of anastomotic leakage is still not completely understood, there is experimental and clinical evidence demonstrating the role of bacteria in the pathogenesis of anastomotic leakage as well as the prevention of anastomotic leakage with the use of topical antibiotics [9,10,11,12,13,14,15,16,17,18,19,20,21,22,23,24,25,26,27,28,29,30,31,32,33].

This is the first study that specifically looked at infectious complications after esophagectomy and anastomotic leakage in a large multicenter cohort. Only patients undergoing Ivor Lewis TMIE were included, allowing for a homogenous study population, thus preventing bias caused by different operation techniques. As totally minimal invasive techniques are increasingly used for the surgical treatment of esophageal cancer, this study will be of use to future surgical practice. We are aware, however, that other techniques exist and evolve and may influence the rate of postoperative complications. SDD may not have the same positive effect in these patient groups, however, the rationale of the prevention of micro-aspiration induced infection and prevention of early micro-abscess formation at the site of anastomosis still stands, irrespective of the technique used.

The main limitation of this study is its failure to fully correct for potential confounding factors because of the retrospective observational design of the study as well as the fact that the SDD strategy in this study depended on the hospital wherein the patient was operated. Nonetheless, the variables that were different between SDD and non-SDD patients did not affect our main findings. There may, however, be residual variables (i.e., differences between hospitals) that we have not accounted for in this study, such as the learning curve or other aspects of perioperative care. To take a closer look at possible confounding factors, an additional sensitivity analysis was performed in two hospitals, showing that only the hospital that did use SDD showed a significant decrease of (infectious) complications over the study period (Figure 1). This analysis reinforces our notion that SDD could indeed be an independent factor responsible for the decreased incidence of pneumonia and anastomotic leakage, irrespective of the potential confounding due to differences between hospitals. Another limitation of this study is that we only had access to the prescription rates of SDD, but not on patient (non)compliance because of possible side effects or intolerance. Moreover, we did not have information on the exact composition and on the validation process of SDD. The SDD paste and suspension are not available in a mass-produced product in the Netherlands and need to be compounded by pharmacy. Therefore, there might be discrepancies regarding formulation and stability in the preparations of SDD in different pharmacies. Lastly, for practical reasons, we did not collect microbiological data.

Some may say that perioperative SDD application is difficult to implement in daily practice. However, similar preoperative prophylactic measures like *Staphylococcus aureus* decolonization with mupirocin for other types of surgery have consistently been shown to be operational. It is true that, in Dutch ICU practice, SDD is generally well accepted as it is part of usual care, facilitating its implementation in everyday practice. Previous studies have shown that this strategy was easily transferred into other healthcare settings in different countries [15,16,17].

In conclusion, our study showed that perioperative SDD use seems to be associated with a lower risk of postoperative pneumonia and anastomotic leakage after Ivor Lewis TMIE. Despite its limitations, this study provides promising results in favor of SDD, which should be confirmed by a Randomized Controlled Trial (RCT) before becoming standard of care in esophagectomy.

## Figures and Tables

**Figure 1 antibiotics-10-00043-f001:**
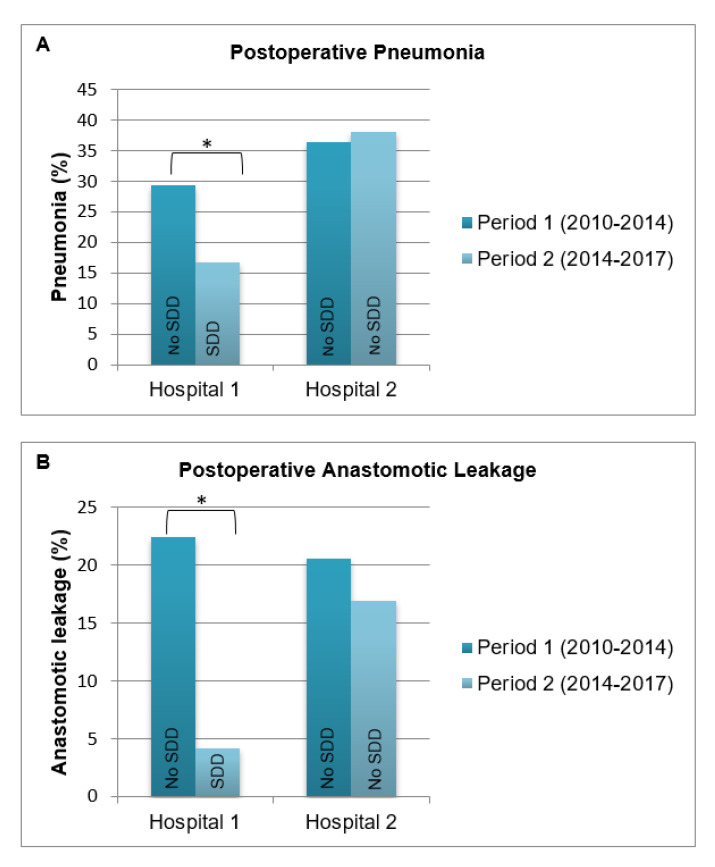
Proportion of pneumonia and anastomotic leakage by time period shown for hospital 1 (N = 180) and hospital 2 (N = 199). * *p*-value of ≤ 0.05.

**Table 1 antibiotics-10-00043-t001:** Baseline characteristics, surgical characteristics, postoperative consequences, and postoperative (infectious) complications, comparing patients using SDD and patients not using SDD.

		Use of SDD		
	Total (n = 496)	Yes (n = 179)	No (n = 317)	*p*-Value
**Patient characteristics, n (%)**				
Male sex	417 (84.1%)	150 (83.8%)	267 (84.2%)	0.900
Age, mean (sd)	64.9 (8.4)	64.6 (9.0)	65.1 (8.1)	0.548
Body mass index, mean (sd)	26.0 (4.1)	25.9 (4.4)	26.0 (3.9)	0.778
ASA classification ^a^				0.004
1	57 (11.5%)	32 (17.9%)	25 (7.9%)	
2	308 (62.1%)	102 (57.0%)	206 (65.2%)	
≥3	130 (26.2%)	45 (25.1%)	85 (26.9%)	
Charlson Comorbidity Index				0.648
0	257 (51.8%)	91 (50.8%)	166 (52.4%)	
1	149 (30.0%)	58 (32.4%)	91 (28.7%)	
≥2	90 (18.1%)	30 (16.8%)	60 (18.9%)	
Neoadjuvant treatment	471 (95.0%)	170 (95.0%)	301 (95.0%)	0.992
Tumor histology ^b^				0.347
Squamous cell carcinoma	57 (11.5%)	24 (13.5%)	33 (11.0%)	
Adenocarcinoma	413 (83.3%)	149 (83.7%)	264 (87.7%)	
Other	9 (1.8%)	5 (2.8%)	4 (1.3%)	
Clinical tumor stage ^c^				<0.001
I	123 (24.8%)	20 (11.4%)	103 (32.8%)	
II	219 (44.2%)	72 (40.9%)	147 (46.8%)	
III	148 (29.8%)	84 (47.7%)	64 (20.4%)	
Tumor location ^d^				0.001
Mid esophagus	28 (5.6%)	11 (6.2%)	17 (5.6%)	
Distal esophagus	343 (69.2%)	142 (80.2%)	201 (66.1%)	
Junction	110 (22.2%)	24 (13.6%)	86 (28.3%)	
**Surgical characteristics**				
Duration of surgery in minutes, median (IQR)	283 (225–355)	294 (209–355)	296 (228–358)	0.587
Epidural analgesia ^e^	475 (95.8%)	176 (98.3%)	299 (94.3%)	0.135
Nr. of resected lymph nodes (IQR)	20 (16–26)	20 (16–23)	23 (16–28)	0.001
Blood loss in ml, median (IQR)	100 (100–200)	200 (20–200)	174 (100–200)	0.042
Conversion to open surgery	20 (4.0%)	15 (8.4%)	5 (1.6%)	<0.001
**Postoperative consequences**				
Hospital readmission	87 (17.5%)	33 (18.4%)	54 (17.0%)	0.694
ICU readmission	83 (16.7%)	27 (15.1%)	56 (17.7%)	0.459
Reintervention	146 (29.4%)	50 (27.9%)	96 (30.3%)	0.581
Length of stay (days)				
Hospital, median (IQR)	12 (9–19)	13 (10–20)	11 (8–19)	<0.001
ICU, median (IQR)	2 (1–4)	4 (2–7)	1 (1–2)	<0.001
**All-cause mortality**				
In hospital	13 (2.6%)	3 (1.7%)	10 (3.2%)	0.393
30-day	11 (2.2%)	3 (1.7%)	8 (2.5%)	0.753
90-day	26 (5.2%)	11 (6.1%)	15 (4.7%)	0.493
1-year ^f^	107 (21.6%)	48 (26.8%)	59 (20.4%)	0.056
**Postoperative complications**				
Pneumonia	153 (30.8%)	36 (20.1%)	117 (36.9%)	<0.001
Surgical-site infections	17 (3.4%)	9 (5.0%)	8 (2.5%)	0.141
Thoracic	0 (0.0%)	0 (0.0%)	0 (0.0%)	
Abdominal	3 (0.6%)	1 (0.6%)	2 (0.6%)	
Jejunostomy	14 (2.8%)	8 (4.5%)	6 (1.9%)	
Anastomotic leakage	82 (16.5%)	19 (10.6%)	63 (19.9%)	0.008
Grade 1	3 (0.6%)	1 (0.6%)	2 (0.6%)	
Grade 2	59 (11.9%)	12 (6.7%)	47 (14.8%)	
Grade 3	20 (4.0%)	6 (3.4%)	14 (4.4%)	
Mediastinitis/empyema	60 (12.1%)	21 (11.7%)	39 (12.3%)	0.851
Chylothorax	48 (9.7%)	23 (12.8%)	25 (7.9%)	0.073


sd, standard deviation; ASA, American Society of Anesthesiology physical status; IQR, interquartile range; Nr., number; ml, milliliter; ICU, intensive care unit; SDD, selective decontamination of the digestive tract. ^a^ Patients from whom ASA score could not be assessed were excluded. Total: 1. ^b^ Patients from whom tumor histology could not be assessed were excluded. Total: 17. ^c^ Patients from whom clinical tumor stage could not be assessed were excluded. Total: 5. ^d^ Patients from whom tumor location could not be assessed were excluded. Total: 15. ^e^ Patients from whom epidural analgesia could not be assessed were excluded. Total: 5. ^f^ Patients from whom 1-year mortality could not yet be assessed were excluded. Total: 37.

**Table 2 antibiotics-10-00043-t002:** Effect of SDD on postoperative (infectious) complications and 1-year mortality.

	Crude		Adjusted ^a^	
	OR (95% CI)	*p*-Value	OR (95% CI)	*p*-Value
Pneumonia	0.43 (0.28–0.66)	<0.001	0.40 (0.23–0.67)	<0.001
Surgical-site infections ^b^	2.05 (0.78–5.40)	0.149	..	..
Anastomotic leakage	0.48 (0.28–0.83)	0.009	0.46 (0.26–0.84)	0.011
1-year mortality	1.53 (0.99–2.38)	0.057	1.31 (0.80–2.14)	0.278

OR, odds ratio; CI, confidence interval. ^a^ Models were adjusted for the following: Pneumonia: body mass index (BMI), Charlson Comorbidity Index, year of surgery, duration of surgery, number of resected lymph nodes, and blood loss. Anastomotic leakage: BMI, year of surgery, and duration of surgery. 1-year mortality: age, Charlson Comorbidity Index, tumor stage, and duration of surgery. ^b^ Surgical-site infections: numbers were too low to perform multivariate analysis.

## Data Availability

The data presented in this study are available on request from the corresponding author. The data are not publicly available due to impracticality of public access.

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
