# Peer review of "Selective Decontamination of the Digestive Tract to Prevent Postoperative Pneumonia and Anastomotic Leakage after Esophagectomy: A Retrospective Cohort Study"

_antibiotics, 2021, doi:10.3390/antibiotics10010043_

Round 1

Reviewer 1 Report

The manuscript submitted by Janssen et al is an important piece of work, and I hope it will reach the targeted audience. I only have a few minor terminological comments and a few questions for authors that will not change my endorsement for publication after they correct a few minor remarks.

Q1: How many surgeons performed these surgeries? Did you find any relationship between post-op complications and surgeons? Were any surgeons involved in both groups (SDD and no SDD)? I know this is a very sensitive topic.

Q2: Do the people in acknowledgement desreve authorship since they provided data?

Minor remarks:

TITLE: Please avoid acronyms in the title.

ABSTRACT: L35-36: It might be useful to specify components parentheses in L30, after SDD.

L39-40: what about the other 317 patients? Please reformulate the sentence.

L42: suggestion: multivariate analysis showed that SDD is an independent protective....

please define OR

L52: please replace Microbial flora with microbiota. Bacteria, archaea and fungi are not plants and have been their own domains/kingdoms for several decades.

L61: anaerobic microbiota instead of flora.

L107-108: what is the carier/solvent in SDD?

L232: microbiota instead of microbiological flora. 

Author Response

Response to Reviewer 1 comments

The manuscript submitted by Janssen et al is an important piece of work, and I hope it will reach the targeted audience. I only have a few minor terminological comments and a few questions for authors that will not change my endorsement for publication after they correct a few minor remarks.

  • RESPONSE: We thank the reviewer for the positive comments. A point-by-point response to the comments and questions follows below.

Q1: How many surgeons performed these surgeries? Did you find any relationship between post-op complications and surgeons? Were any surgeons involved in both groups (SDD and no SDD)? I know this is a very sensitive topic.

  • RESPONSE: These are relevant questions, which we will answer one by one:
  • All the operations were performed in teams of two surgeons. The number of surgeons per hospital specially trained to perform this specific operation differed per hospital: in hospital 1, there were three surgeons, in hospital 2 there were two surgeons, in hospital 3 there were also two surgeons, and in hospital 4 there were three surgeons. Most of the time, the qualified surgeons operated together, however, sometimes the surgery was performed by one specially trained surgeon in combination with another general GI-surgeon or a surgeon in training. We added this to the method section (lines 94-100).
  • The question of the reviewer asking if we found any relationship between post-operative complications and surgeons is relevant. We indeed tried to look at this association, however, as the surgical teams, even within the hospital, were not always exactly the same and it was unclear who was the true lead surgeon when operating together, it was unfortunately not possible to assess a relationship for each individual surgeon. We decided that, as these operations were done by compact teams of dedicated surgeons, an evaluation of individual surgeons rather than teams was not feasible.
  • None of the surgeons were involved in both groups (SDD and no SDD), as they all worked at only one hospital.

Q2: Do the people in acknowledgement deserve authorship since they provided data?

  • RESPONSE: Thank you for mentioning this. We now realize that this needs some clarification. The persons mentioned in the acknowledgements did not directly provide us the data, they provided us access to the data/patient-files after, of course, approval of the ethical committee and local hospital committee (i.e. they were the contact persons from the other hospitals). As they were not involved in the study design, data collection, data analysis, writing of the manuscript or in any other task, their role was insufficient for a full authorship. Based on the comment, we adjusted the acknowledgements (lines 306-307).

Minor remarks:

TITLE: Please avoid acronyms in the title.

  • RESPONSE: Thank you for pointing this out. We removed, as suggested, the acronym “SDD” from the title.

ABSTRACT: L35-36: It might be useful to specify components parentheses in L30, after SDD.

  • RESPONSE: As suggested by the reviewer, we placed the components of SDD in parentheses (line 36).

L39-40: what about the other 317 patients? Please reformulate the sentence.

  • RESPONSE: Thank you for pointing this out. We added a part to the sentence stating that the other 317 patients did not receive SDD (lines 40-41).

L42: suggestion: multivariate analysis showed that SDD is an independent protective....

please define OR

  • RESPONSE: Thank you for your suggestion, we changed this sentence accordingly (line 42-43). We also defined Odds Ratio (OR) in de abstract (line 39) and in the methods (lines 154-155).

L52: please replace Microbial flora with microbiota. Bacteria, archaea and fungi are not plants and have been their own domains/kingdoms for several decades.

  • RESPONSE: We thank the reviewer for pointing this out. As suggested, we changed “microbial flora” to “microbiota” (line 53).

L61: anaerobic microbiota instead of flora.

  • RESPONSE: We replaced “flora” with “anaerobic microbiota” (line 62)

L107-108: what is the carier/solvent in SDD?

  • RESPONSE: The base of SDD consists of vaselinum album, paraffin and methocel, and it is slightly sweetened and has peppermint taste. We added the complete ingredient list as an example (this might slightly different between hospitals).

L232: microbiota instead of microbiological flora.

  • RESPONSE: As suggested by the reviewer, we changed “microbiological flora” to “microbiota” (line 239)

Reviewer 2 Report

This is a retrospective cohort study that determined the effects of selective decontamination of the upper digestive tract on incidence of pneumonia and anastomotic leak in patients undergoing Ivor-Lewis esophagectomy. This manuscript is well written, the limitations are well described, and recommendations for future prospective studies are forwarded. The costs of post-operative pneumonia and anastomotic leak are estimated to be at least $15,000 and $68,000 USD, respectively. Because the study showed no differences in mortality, but did illustrate increased length of stay in hospital and ICU for the SDD group, future studies need to validate the absolute reduction in morbidity and estimate resource utilization impacts from avoiding pneumonia and leaks.

One area of the paper that could be strengthened is a fuller and cited description of the SDD agents. The paste and suspension are not available in a mass produced product in the Netherlands, and need to be compounded by pharmacy. There was no validation process described, and there may be discrepancies regarding formulation and stability in preparations from different pharmacies. Therefore, including several citations about how the paste and suspension were compounded and supplied, both for in-patient and out-patient administration, would strengthen your arguments. If it is not known, you should include in it the limitations section. 

Otherwise,

title is descriptive, abstract is summative, and key words are appropriate. Tables and figures are very useful and illustrative.

References are not in mdpi style and need to be modified.

Excellent work that provides more clarity for current practice and future bundle guideline development.

Author Response

Response to Reviewer 2 comments

This is a retrospective cohort study that determined the effects of selective decontamination of the upper digestive tract on incidence of pneumonia and anastomotic leak in patients undergoing Ivor-Lewis esophagectomy. This manuscript is well written, the limitations are well described, and recommendations for future prospective studies are forwarded. The costs of post-operative pneumonia and anastomotic leak are estimated to be at least $15,000 and $68,000 USD, respectively. Because the study showed no differences in mortality, but did illustrate increased length of stay in hospital and ICU for the SDD group, future studies need to validate the absolute reduction in morbidity and estimate resource utilization impacts from avoiding pneumonia and leaks.

One area of the paper that could be strengthened is a fuller and cited description of the SDD agents. The paste and suspension are not available in a mass produced product in the Netherlands, and need to be compounded by pharmacy. There was no validation process described, and there may be discrepancies regarding formulation and stability in preparations from different pharmacies. Therefore, including several citations about how the paste and suspension were compounded and supplied, both for in-patient and out-patient administration, would strengthen your arguments. If it is not known, you should include in it the limitations section.

  • RESPONSE: The reviewer is right in mentioning that paste and suspension are not available in a mass produced product in the Netherlands and that there may be slight differences in the preparations form the different pharmacies. So -in theory- these differences may have affected the outcomes of our study. However, even if the manufacturing process of SDD has not been described in national guidelines, all hospitals have used SDD paste and suspension containing amphotericin B, tobramycin and colistin, so the coverage of microbiota was exactly the same. We indeed have not been able to establish the exact composition in all different study hospitals so we will add this as a limitation to the study in the discussion section (lines 274-278). As for the difference between out and inpatient SDD: in these hospitals SDD was provided from the hospital pharmacy where the patient was operated.

Otherwise, title is descriptive, abstract is summative, and key words are appropriate. Tables and figures are very useful and illustrative.

  • RESPONSE: We are grateful for the positive comments.

References are not in mdpi style and need to be modified.

  • RESPONSE: Thank you for pointing this out. We changed, as suggested, the references to the MDPI style (page 13-14).

Excellent work that provides more clarity for current practice and future bundle guideline development.

  • RESPONSE: We thank the reviewer once again for the compliments.